# Drivers of Under-Five Stunting Trend in 14 Low- and Middle-Income Countries since the Turn of the Millennium: A Multilevel Pooled Analysis of 50 Demographic and Health Surveys

**DOI:** 10.3390/nu11102485

**Published:** 2019-10-16

**Authors:** Alemayehu Argaw, Giles Hanley-Cook, Nathalie De Cock, Patrick Kolsteren, Lieven Huybregts, Carl Lachat

**Affiliations:** 1Department of Food Technology, Safety and Health, Faculty of Bioscience Engineering, Ghent University, 9000 Ghent, Belgium; Giles.HanleyCook@ugent.be (G.H.-C.); nadecock@gmail.com (N.D.C.); patrick.kolsteren@ugent.be (P.K.); carl.lachat@ugent.be (C.L.); 2Department of Population and Family Health, Institute of Health, Jimma University, P.O. Box 378, Jimma, Ethiopia; 3Poverty, Health and Nutrition Division, International Food Policy Research Institute, Washington, DC 20005-3915, USA; l.huybregts@cgiar.org

**Keywords:** stunting trend, children under-five years, low- and middle-income countries, determinants, Demographic and Health Survey (DHS)

## Abstract

Background: Understanding the drivers contributing to the decreasing trend in stunting is paramount to meeting the World Health Assembly’s global target of 40% stunting reduction by 2025. Methods: We pooled data from 50 Demographic and Health Surveys since 2000 in 14 countries to examine the relationships between the stunting trend and potential factors at distal, intermediate, and proximal levels. A multilevel pooled trend analysis was used to estimate the association between the change in potential drivers at a country level and stunting probability for an individual child while adjusting for time trends and child-level covariates. A four-level mixed-effects linear probability regression model was fitted, accounting for the clustering of data by sampling clusters, survey-rounds, and countries. Results: Stunting followed a decreasing trend in all countries at an average annual rate of 1.04 percentage points. Among the distal factors assessed, a decrease in the Gini coefficient, an improvement in women’s decision-making, and an increase in urbanization were significantly associated with a lower probability of stunting within a country. Improvements in households’ access to improved sanitation facilities and drinking water sources, and children’s access to basic vaccinations were the important intermediate service-related drivers, whereas improvements in early initiation of breastfeeding and a decrease in the prevalence of low birthweight were the important proximal drivers. Conclusions: The results reinforce the need for a combination of nutrition-sensitive and -specific interventions to tackle the problem of stunting. The identified drivers help to guide global efforts to further accelerate stunting reduction and monitor progress against chronic childhood undernutrition.

## 1. Introduction

Stunting occurs in more than one in five children under five years, affecting 149 million globally [1]. Childhood stunting is strongly associated with an array of short- and long-term health and economic consequences, including greater risks of childhood infections, developmental deficits resulting in poorer school performance and lower productivity and earnings in adulthood, and early physiological changes contributing to an increased risk of adult noncommunicable diseases and mortality [2,3,4]. Stunting also leads to undesired outcomes in the following generation, where newborns from stunted mothers are more likely to be small-for-gestational-age with an increased risk of later growth faltering and morbidity and mortality [5].

In recognition of stunting as an important outcome in itself and as a marker of several adverse outcomes, global attention has been directed towards reducing the high burden of childhood stunting in low- and middle-income countries (LMICs) [6]. In 2012, the World Health Assembly endorsed global nutrition targets for 2025, including a 40% reduction in the number of children under five years old who are stunted [7]. While stunting has declined over the last three decades, the current trend in many LMICs is insufficient to reach the target for 2025 [1]. To reach this target, it is essential to understand the important drivers behind the observed trend in stunting reductions in these countries.

While a great number of studies relied on cross-sectional variations to explore the determinants of childhood stunting [8,9,10,11,12,13,14], only a few tried to understand which of these time-variant factors influenced the trend in stunting prevalence observed over time [15,16,17,18,19,20]. Moreover, most efforts to assess stunting reduction trends have taken a country focus and examined contextual drivers [15,16,17,18,19,20]. However, comprehensive evidence of stunting drivers across high burden countries is equally important for decision-making at the global level. Only a few studies have used a cross-country analysis to explore the association between stunting trends and potential drivers [21,22,23,24,25]. However, previous analyses were limited due to the unavailability of data on important nutrition-related determinants. For instance, the use of per capita energy supply may be a distant proxy for the underlying infant and young child feeding (IYCF) practice indicators [21,22,24]. Furthermore, timely revision of evidence on the drivers of the current stunting trend is relevant for decision-making as the strength of the relationship between stunting and potential drivers can change over time [16,17].

In this study, we used a large cross-country dataset of the Demographic and Health Surveys (DHS) to understand the association between changes in a range of distal to proximal factors and stunting prevalence among under-five children in 14 LMICs since 2000.

## 2. Materials and Methods

### 2.1. Data Source

We considered data from DHSs conducted in 42 LMICs countries that are partners for nutrition support by the European Commission’s Directorate-General for International Cooperation and Development. Our trend analysis required data from at least 3 time-points. Therefore, the present analysis only utilized data from 14 countries with an adequate number of standard DHS rounds since 2000. Countries included in our analysis were Bangladesh, Cambodia, Ethiopia, Haiti, Kenya, Malawi, Mali, Nepal, Nigeria, Rwanda, Tanzania, Uganda, Zambia, and Zimbabwe.

The DHS has been conducted in several countries for over three decades, providing nationally representative cross-sectional data on demographic, health, and nutrition information, among others [26]. DHS uses standardized data collection procedures across countries and consistent content across rounds, allowing for comparison of data across-countries and within-country over time. We pooled data from 50 DHS rounds in the 14 countries. There were a different number of surveys per country and different time spans between surveys for the countries. Data were compiled from the official website of the DHS Program (https://dhsprogram.com/) accessed with permission in November 2018. Children under five years of age with height measurements and other relevant information available were included in the analysis.

### 2.2. Variables

The outcome of interest was the probability of being stunted in under-five children. Stunting here is described as binary (stunted/not stunted), defined as a height/length-for-age *z* score below 2 SD from the median based on the WHO 2006 Child Growth Standards [27]. Potential drivers for stunting reduction were considered based on UNICEF’s conceptual framework for the determinants of childhood undernutrition [28] and the Lancet review for the framework of action [29]. A variable was selected for analysis when it was available in the DHS datasets for enough survey rounds, used as a determinant in more than one paper, or mentioned in reviews and/or were found to be significant in previous country analyses. We considered three groups of variables operating at different levels as distal factors, intermediate health and related services utilization factors, and proximal factors (Table 1) [25]. We considered variables like education coverage and women’s decision-making power and work opportunity as distal factors because these factors can influence access to and utilization of the intermediate service-related variables. Definitions of our potential drivers are in line with the DHS statistics guideline [30]. Additionally, the associations between drivers and stunting were adjusted for important demographic and socioeconomic covariates for the child, mother, and household, including child age, sex, birth-order and birth-interval, maternal age and marital status, household wealth status, and place of residence (urban/rural).

### 2.3. Statistical Analysis

Data management and statistical analyses were conducted in Stata version 14.1 (StataCorp LLC, College Station, TX, USA) using the High Performance Computing infrastructure at Ghent University. For all analyses, associations with a *p*-value of less than 0.05 were regarded as statistically significant. Missing indicators for certain survey-rounds were imputed based on linear interpolation of the available time-points within a country. The weighted prevalence and average values of stunting and explanatory indicators in each survey-round were calculated considering the DHS sampling weight factor using the *svyset* command. Descriptive evaluation of the trend in stunting and the explanatory indicators over survey-rounds within-country was conducted using locally weighted scatterplot smoothing graphs. The pooled annualized rate of change in stunting prevalence and explanatory variables were estimated by fitting mixed-effects linear regression models with a random intercept country and as a random slope survey round, and applying weightings based on the population size of each country.

We followed the approach by Fairbrother [31] to analyze repeated cross-sectional survey datasets from different subjects. The method allows for using individual-subject level data to examine the association between aggregate level data on potential drivers and stunting risk for an individual child, while adjusting for time trends (survey year) and important demographic and socioeconomic covariates for the child, mother, and household mentioned above. A mixed-effects linear probability model with a robust variance estimator was fitted using four-level random intercepts to account for potential sources of clustering in our data where individuals (level-1) are nested within the DHS sampling clusters (level-2), which in turn are nested within survey-rounds (level-3), and finally, survey-rounds nested within countries (level-4). Since an indicator can vary both within-country, over time, and across-country, we sought to identify separate within-country (longitudinal) and between-country (cross-sectional) components for the association between an indicator variable and stunting. For this purpose, for each indicator variable, we calculated the average value per country (representing cross-country differences) and the average value per survey-round in a country mean-centered (representing within-country change over time), which were simultaneously specified in the model. The within-between model specification has the additional advantage of minimizing potential endogeneity problems in mixed-effects models [32]. In order to avoid over-adjustment from the inclusion of distal, intermediate, and proximal indicators in a single model, we specified three separate multivariable regression models for each group of indicators. The relationships between an explanatory variable and stunting in the four-level models are represented using the equation below:

y_iktj_ = β_0_ + β_1_x_tjM_ + β_2_x̅_j_ + β_3_x_iktj_ + β_4_time_tj_ + u_k_ + u_kj_ + u_ktj_ + ε_iktj_;

where y_iktj_ is the risk of stunting in child i who is from a DHS sampling cluster k nested within a country-year tj, which in turn is nested within a country j. β_1_ and β_2_ give estimates for stunting-indicator associations decomposed into between-country (which is estimated from the value of an indicator at a country level using the average of all survey-rounds in a country (x̅_j_)) and within-country effects (which is estimated from the average value of an indicator for each survey-round per country (x_tjM_)), respectively. Country-year level variables were mean-centered by subtracting the value of the country-level variable from the value at survey-round within the same country. β_3_ gives the estimate for the association of individual-subject level covariates used for adjustment in the model. β_4_ gives the estimate for the time variable as a set of year dummies, which adjust for possible simultaneous but unrelated time trends in both x_j_/x_tjM_ and stunting. Random effect components modeling clustering of data by the DHS sampling cluster (u_k_), by country-years (u_kj_), and countries (u_ktj_), and the residual term for individual child (ε_iktj_) are assumed to follow a Gaussian distribution with a mean value of zero.

## 3. Results

Our final dataset included 322,320 children under-five from 50 survey-rounds in 14 countries that were conducted between 2000 and 2016 (Table 2). The mean (SD) child age was 28.6 (17.2) months and 49.8% of children were female. The mean (SD) maternal age was 28.7 (6.86) years. The number of survey-rounds per country were three in six countries and four in eight countries. The average sample size per survey-round and per country was 6457 (range: 2070–24,505).

The trend over survey-rounds for stunting prevalence and the indicator variables within the 14 study countries are presented in Figure 1 and Appendix A. Stunting prevalence shows a declining trend over time in all countries assessed. The overall average annual rate of stunting reduction was 1.04 percentage points (pp) (β (95% CI): −1.04 (−1.24, −0.84); *p* < 0.001) with differing slopes among countries (Table 3). Similarly, most of the indicator variables also showed improvements over time except in some variables, like coverage of women’s work opportunity.

We estimated the association between the probability of stunting and the variations in potential drivers within- and between-countries with the within-country association reported as a measure of the effect of an indicator on stunting trend over time (Table 4). The Gini coefficient, urbanization, and women’s decision-making power were the distal-level indicators found to be significantly associated with change in stunting risk within a country. An SD increase in the Gini coefficient was associated with an increased probability of stunting by 1.10 pp (*p* = 0.008). A 10% increase in the percentage of the population living in urban areas and women with decision-making power was associated with a 0.67 pp (*p* = 0.013) and 0.36 pp (*p* = 0.040) decreased the probability of stunting, respectively.

Households’ access to improved sanitation facilities and drinking water sources, and children’s access to all basic vaccinations were among the intermediate level health and related service determinants found to be significantly associated with stunting risk overtime. Within a country, a 10% increase in the coverage of improved sanitation facilities, improved drinking water sources, and basic child vaccinations were associated with decreases in the probability of stunting by 1.40 pp (*p* < 0.001), 1.48 pp (*p* = 0.003), and 1.74 pp (*p* < 0.001), respectively.

As for the proximal diet and health related factors, prevalence of reported low birthweight and percent of children who initiated breastfeeding within one day of birth were significantly associated with changes in stunting risk overtime. A 10% decrease in the prevalence of low birthweight over time was associated with a 3.91 pp (*p* < 0.001) decreased probability of stunting. A 10% increase in children with early initiation of breastfeeding within a country was associated with a 1.17 pp (*p* < 0.001) decrease in the probability of stunting.

## 4. Discussion

This study examined a range of distal to proximal factors that might explain the observed trend in stunting reduction since the turn of the millennium in 14 LMICs. Stunting prevalence has shown a declining trend in all countries assessed, with an average annual reduction rate of 1.04 pp. The key indicators that explained the stunting trend over time were changes in income inequality, urbanization, and women’s decision-making power from the distal factors assessed; changes in household access to improved sanitation facilities and improved drinking water sources, and child immunization rate for basic vaccinations from the intermediate service related factors; and changes in the prevalence of reported low birthweight and early initiation of breastfeeding from the proximal factors.

The current study suggests that investments narrowing economic inequalities among households and empowering women are key drivers to reduce stunting risk in the 14 study countries. In a previous cross-country study of stunting trend from 1970–2001, Milman et al. [22] reported that countries with more equitable income distribution achieved better stunting reduction. Others also reported increased household asset accumulation as an important driver for stunting reduction in both a cross-country study [21], as well as in trend analyses of countries with success stories in stunting reduction [18,19,20]. Our finding that improvements in women’s decision-making power is a key driver of stunting trend is consistent with a review of literature by Carlson et al. [33], which concluded that increases in maternal autonomy were associated with better outcomes of child nutritional status in LMICs. It is widely acknowledged that mothers play a vital role as a primary caregiver of the child, and their control over household decision-making is expected to facilitate the level of childcare and feeding and utilization of health care services that are important underlying factors for child nutritional status and health. On the other hand, women’s education, which has been reported as a key driver of stunting reduction over the last three or more decades in previous trend analyses [18,19,20,21,22,25], was not found to be important in our analysis. Although countries with better coverage of women’s primary education had significantly lower stunting burden, the within-country association was not significant. This suggests that although women’s education has been responsible for stunting changes that occurred over the long-term, it contributed less to the more recent changes over the last two decades. This may be due to a lack of substantial gains in women’s education coverage in the more recent period in the studied countries, which was indicated by the small nonsignificant change observed in this indicator compared to most other indicators in our data (Table 3). Another possible explanation may be that the potential impact of women’s education on stunting trend may have been explained by other closely related variables in our model, such as women’s decision-making power. However, the within-country association for women’s education remains nonsignificant after excluding women’s decision-making power and work opportunity from our model, suggesting that our previous explanation holds.

This study found that within-country changes in stunting risk were associated with improvements in household access to improved sanitation facilities and improved drinking water sources and increased coverage of child immunization, which is consistent with findings from other trend analysis studies [17,18,19,20,22,25]. These findings are supported by the growing body of literature indicating that enteric dysfunction, resulting from poor environmental conditions (water, sanitation, and hygiene), is among the most important factors for the high childhood stunting burden in low-income settings [4,34,35].

The current analysis extends previous trend analysis studies by including the more direct measures of the immediate factors for child undernutrition, including the prevalence of reported low birthweight, IYCF indicators, and prevalence of childhood illnesses. Low birthweight is an important driver of the change in stunting prevalence, which was anticipated as previous estimates showed intrauterine growth restriction (estimated by rates of low birth weight) accounts for 20% of childhood stunting occurring in LMICs [36]. On the other hand, from the IYCF indicators assessed, only improvements in early initiation of breastfeeding were found to be significantly associated with a reduction in the probability of stunting within the 14 countries. We did not find a statistically significant association between stunting and the percentage of breastfed children 6–8 months who started complementary feeding or the median duration of breastfeeding. A possible explanation might be that the mere introduction of solid or semi-solid foods may not be a good indicator of children’s dietary quality. In this regard, indicators such as dietary diversity and meal frequency, and the consumption of animal source foods may have been better indicators to evaluate dietary quality and its association with child linear growth [37,38,39,40,41,42,43]. However, such indicators are available only in the most recent DHS survey rounds. The lack of association between longer duration of breastfeeding and reduction in stunting prevalence has also been reported previously. It is possible that higher reliance on breastfeeding could occur in lower income settings where there is less access to diverse diets for infants, as well as mothers of infants with poor health and growth may decide to continue breastfeeding for a longer duration [44,45,46].

One important limitation of the current study is that the generalizability of findings is limited as our analysis did not include all relevant countries, such as India, which holds almost a third of the world’s stunting burden [1]. Given the cross-sectional nature of the study, we cannot assure observed associations are causal. Furthermore, it was not possible to consider the time-lag between indicators and stunting risk as our analysis was entirely based on DHS datasets, where the predictors and the outcome were collected at the same time. However, experimental impact evaluations, such as controlled interventions, are extremely challenging, if not impossible, to address such holistic research questions addressing multiple drivers of chronic child malnutrition at the level of multiple countries. It should be noted that the indicators that were found to be important drivers in the past may not necessarily be the ones that will be important in the future due to possible changes in indicator-stunting associations and a saturation of some indictors for future improvements. For this purpose, unlike most previous trend analysis studies, we aimed at evaluating a more recent trend since the year 2000. We note that most of the indicators found to be important in our analysis were also targets of the United Nations’ Millennium Development Goals set for 2015 and its continuation of the Sustainable Development Goals for the year 2030 [47]. Thus, it is plausible that with future investment, substantial progress can be made on these indicators with a potential resultant effect on stunting reduction. Finally, some of the potentially relevant factors like complementary feeding, which were not found to be significantly associated with stunting reduction in the current analysis, could be due to lack of improvement in status over time in most of the countries studied. Therefore, we cannot exclude the potential for some of the nonsignificant indicators to contribute to stunting reduction with future investment improving their status in a population.

## 5. Conclusions

In conclusion, the current study identified important distal to proximal potential drivers for the observed trend in stunting reduction, which is useful information for future efforts to further accelerate stunting reduction and monitor progress against chronic childhood undernutrition. Our findings indicate that stunting drivers were present at both distal and intermediate, as well as proximal levels, suggesting that economic development and nutrition-sensitive interventions, on top of nutrition-specific programs, could play an important role in reducing stunting prevalence in these countries.

## Figures and Tables

**Figure 1 nutrients-11-02485-f001:**
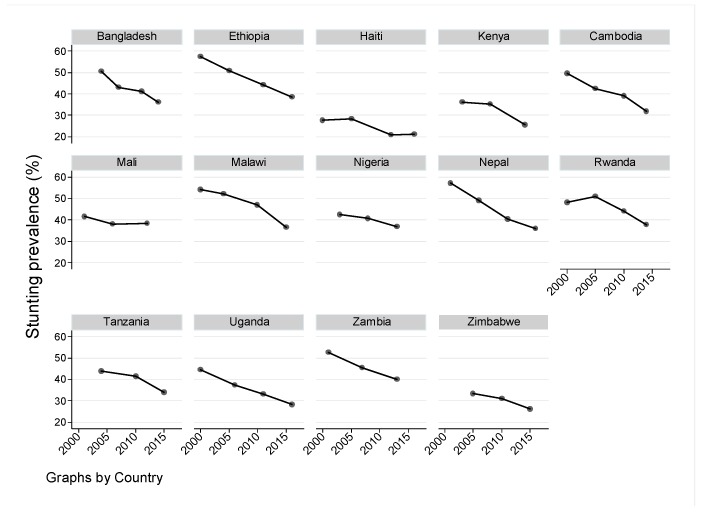
Trends in under-five stunting prevalence by country.

**Table 1 nutrients-11-02485-t001:** Description of variables used in the study ^1^.

Variables	Definition
**Outcome**	
Stunting	Height/length-for-age *z* score value 2.0 SD below the median based on the WHO 2006 Child Growth Standards.
**Distal Determinants**	
Gini coefficient	A coefficient calculated from the DHS household wealth index score indicating the level of concentration of wealth among households in a country. Possible values range from 0 (being an equal distribution) to 1 (a totally unequal distribution) with higher values indicating more unequal distribution with a lower proportion of households controlling more of the wealth in the country.
Total fertility rate	The average number of births women (15–49 years) would have by the time they reach age 50 years based on the current age-specific fertility rate during the 3 years preceding the survey (excluding 1–36 months before the survey).
Urbanization	Percentage of total population living in urban areas.
Female primary education coverage	Percentage of women (15–49 years) who attended any level of primary education.
Male secondary education coverage	Percentage of men (15–49 years) who attended any level of secondary education.
Women decision making power	Percentage of currently married women (15–49 years) who usually make final decisions (alone or jointly with their husband) about their own healthcare, large household purchases, and visits to family or relatives.
Women work opportunity	Percentage of women (15–49 years) who worked in the past 7 days (including women who did not work in the past 7 days, but who are regularly employed and were absent from work for leave, illness, vacation, or any other reason) or worked in the past 12 months, but not currently.
**Intermediate Health and Related Service Determinants**	
Improved sanitation facility coverage	Percentage of total households with improved sanitation following the WHO/UNICEF Joint Monitoring Programme (JMP) for Water Supply, Sanitation and Hygiene.
Improved drinking water sources coverage	Percentage of total households with improved drinking water source following the WHO/UNICEF Joint Monitoring Programme (JMP) for Water Supply, Sanitation and Hygiene.
Antenatal care follow-up with ≥4 visits coverage	Percentage of women with a live birth in the 5 years preceding the survey who received ≥ 4 antenatal care visits for the most recent birth.
Delivery at health facility coverage	Percentage of live births to interviewed women in the 5 years preceding the survey delivered in a health facility.
Iron supplementation during pregnancy coverage	Percentage of women with a birth in the 5 years preceding the survey who took iron tablets or syrup (given or bought) during the pregnancy for the most recent live birth.
Children with all 8 basic vaccinations coverage	Percentage of living children (12–35 months) who received BCG, 3 doses of DPT-containing vaccine, 3 doses of polio vaccine (excluding polio vaccine given at birth), and 1 dose of MCV at any time, according to vaccination card or mother’s report.
**Proximal Determinants**	
Initiation of breastfeeding in ≤1 day	Percentage of last-born children who were born in the 2 years preceding the survey put to the breast within one day of birth.
Median duration of exclusive breastfeeding	Median duration (in months) of exclusive breastfeeding among children born in the past three years.
Complementary feeding between ages 6–9 months	Percentage of children (6–8 months) who were both breastfed and received complementary food (solid or semi-solid foods) in the 24 h preceding the interview.
Prevalence of reported low birthweight	Percentage of live births to interviewed women in the 5 years preceding the survey where the mother’s estimated baby’s size at time of birth as smaller than average.
Prevalence of acute respiratory infection (ARI)	Percentage of living children (0–59 months) with symptoms of ARI at any time in the 2 weeks preceding the survey.
Prevalence of diarrhea	Percentage of living children (0–59 months) with diarrhea at any time in the 2 weeks preceding the survey.

^1^ BCG, Bacille Calmette Guerin vaccine against tuberculosis; DPT, diphtheria-pertussis-tetanus vaccine; DHS, Demographic and Health Survey; MCV, Measles antigen-containing vaccine; SD, standard deviation; UNICEF, United Nations Children’s Fund; WHO, World Health Organization.

**Table 2 nutrients-11-02485-t002:** Characteristics of included Demographic and Health Surveys.

Characteristics		Statistics
Surveys, *n*		50
Countries, *n*		14
Surveys per country, *n*		
	3 rounds	6
	4 rounds	8
Countries with survey years		
	Bangladesh	2004, 2007, 2011, 2014
	Cambodia	2000, 2005, 2010, 2014
	Ethiopia	2000, 2005, 2011,2016
	Haiti	2000, 2005, 2012, 2016
	Kenya	2003, 2008, 2014
	Malawi	2000, 2004, 2010, 2015
	Mali	2001, 2006, 2012
	Nepal	2001, 2006, 2011, 2016
	Nigeria	2003, 2008, 2013
	Rwanda	2000, 2005, 2010, 2014
	Tanzania	2004, 2010, 2015
	Uganda	2000, 2006, 2011, 2016
	Zambia	2001, 2007, 2013
	Zimbabwe	2005, 2010, 2015
Total sample size, *n*		322,320
Children per survey, mean (range)		6457 (2070–24,505)
Child age (months), mean (SD)		28.6 (17.2)
Child sex (female), %		49.8
Maternal age (years), mean (SD)		28.7 (6.86)

**Table 3 nutrients-11-02485-t003:** Trends in stunting prevalence and potential determinants across survey-rounds (*n* = 50) ^1.^

Variables	Average Annualized Rate of Change	
	Beta (95% CI)	*p*
Stunting prevalence in under-five children (%)	−1.04 (−1.24, −0.84)	<0.001
Gini coefficient	−0.01 (−0.02, −0.01)	<0.001
Total fertility rate (average number of births per women)	−0.07 (−0.09, −0.05)	<0.001
Urbanization (%)	0.61 (0.21, 1.00)	0.003
Female primary education (%)	0.21 (−0.49, 0.91)	0.560
Male secondary education (%)	0.13 (−0.05, 0.30)	0.163
Women’s decision-making power (%)	2.31 (1.36, 3.26)	<0.001
Women working (%)	−0.30 (−0.84, 0.25)	0.286
Improved sanitation facilities (%)	1.87 (1.02, 2.72)	<0.001
Improved drinking water sources (%)	1.25 (0.66, 1.84)	<0.001
Antenatal care follow-up with ≥4 visits (%)	1.13 (0.53, 1.72)	<0.001
Delivery at health facility (%)	1.82 (1.17, 2.47)	<0.001
Iron supplementation during pregnancy (%)	0.39 (−2.59, 3.37)	0.798
Children with all 8 basic vaccinations (%)	1.18 (0.98, 1.38)	<0.001
Initiation of breastfeeding in ≤1 day (%)	1.00 (0.66, 1.33)	<0.001
Median duration of exclusive breastfeeding (months)	0.08 (0.04, 0.12)	<0.001
Complementary feeding between ages 6–9 months (%)	0.32 (−0.21, 0.86)	0.239
Prevalence of reported low birthweight (%)	−0.18 (−0.58, 0.23)	0.392
Prevalence of acute respiratory illness (%)	−1.00 (−1.14, −0.86)	<0.001
Prevalence of diarrhea (%)	−0.47 (−0.69, −0.26)	<0.001

^1^ Average annualized rates of change (Beta coefficients (95% CI)) were estimated using a mixed-effects linear model with country as random intercept and time as random slope and using weighting for each country’s population size.

**Table 4 nutrients-11-02485-t004:** Modeling distal, intermediate, and proximal drivers of trend in stunting prevalence ^1^.

Indicators	Stunting	
	Beta (95% CI)	*p*
**Distal determinants**		
Gini coefficient (SD)	**1.10 (0.29, 1.91)**	**0.008**
Total fertility rate (SD)	0.39 (−0.78, 1.56)	0.513
Urbanization (10%)	**−0.67 (−1.21, −0.14)**	**0.013**
Women primary education (10%)	−1.12 (−2.42, 0.18)	0.092
Male secondary education (10%)	−1.95 (−5.47, 1.56)	0.275
Women’s decision making (10%)	**−0.36 (−0.69, −0.02)**	**0.040**
Women’s working (10%)	−0.72 (−2.58, 1.15)	0.450
**Intermediate service related determinants**		
Improved sanitation facilities (10%)	**−1.40 (−2.17, −0.62)**	**<0.001**
Improved drinking water sources (10%)	**−1.48 (−2.45, −0.52)**	**0.003**
Antenatal care follow-up with ≥4 visits (10%)	0.38 (−0.46, 1.23)	0.377
Delivery at health facility (10%)	0.08 (−1.20, 1.36)	0.907
Iron supplementation during pregnancy (10%)	0.21 (−0.14, 0.56)	0.240
Children with all 8 basic vaccinations (10%)	**−1.74 (−2.53, −0.94)**	**<0.001**
**Proximal determinants**		
Initiation of breastfeeding in ≤1 day (10%)	**−1.17 (−1.62, −0.73)**	**<0.001**
Median duration of exclusive breastfeeding (SD)	0.18 (−0.32, 0.68)	0.473
Complementary feeding b/n age 6–9 months (10%)	0.75 (−0.53, 2.03)	0.250
Prevalence of reported low birthweight (10%)	**3.91 (2.40, 5.43)**	**<0.001**
Prevalence of acute respiratory illness (10%)	1.28 (−1.40, 3.96)	0.349
Prevalence of diarrhea (10%)	−2.73 (−6.12, 0.65)	0.113

^1^ Beta coefficients (95% CI) are estimated using a mixed-effects linear probability regression model with a robust variance estimator and using four-level random intercept accounting for clustering of individuals by sampling clusters, survey-rounds, and countries. Separate models were fitted for each group of the distal, intermediate, and proximal variables. Models were adjusted for time trend and important individual-level covariates including child age, sex, birth-order and birth-interval, maternal age and marital status, household wealth status, and place of residence (urban/rural). Bold values indicate *p* < 0.05. SD, standard deviation.

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
