# Peer review of "Drivers of Under-Five Stunting Trend in 14 Low- and Middle-Income Countries since the Turn of the Millennium: A Multilevel Pooled Analysis of 50 Demographic and Health Surveys"

_nutrients, 2019, doi:10.3390/nu11102485_

Round 1

Reviewer 1 Report

Line 209: drivers rather than divers

Line 216: were (increases were associated with better outcomes...)

Line 234: were (changes were associated with improvements...)

Line 247: were (improvements were found...)

Author Response

Line 209: drivers rather than divers

Response: Corrected 

Line 216: were (increases were associated with better outcomes...)

Response: Corrected 

Line 234: were (changes were associated with improvements...)

Response: Corrected 

Line 247: were (improvements were found...)

Response: Corrected 

Reviewer 2 Report

Dear authors, 

Indeed a nice piece of work. Comprehensive use of available data, use of an underlying conceptual (Unicef) model guiding the analyses and use of mixed-effects statistical models to analyse the predictors of change in prevalence of child stunting. 

While I am have limited experience of mixed effects models I wonder whether aggregation of individual data to become higher level data makes the most efficient use of DHS data. It would be good to see this discussed as well as the potential pitfalls of use of aggregated data in mixed-effects models. 

As a matter of rigour, while the use of "drivers" raises more interest it has a connotation of causality which this cross-sectional data cannot support. You have discussed this as a limitation and used the term "predictors". To be humble and acknowledge this limitation I suggest that you change the title and relevant wording in the text.   

Author Response

Dear authors, 

Indeed a nice piece of work. Comprehensive use of available data, use of an underlying conceptual (Unicef) model guiding the analyses and use of mixed-effects statistical models to analyse the predictors of change in prevalence of child stunting. 

While I am have limited experience of mixed effects models I wonder whether aggregation of individual data to become higher level data makes the most efficient use of DHS data. It would be good to see this discussed as well as the potential pitfalls of use of aggregated data in mixed-effects models. 

Response: The use of an aggregate data for the predictors is by design. Our research question is to assess the association between change over time in the drivers and the outcome. We are applying this research question on repeated cross-sectional survey dataset from different subjects. Since we don’t have longitudinal data per subject (repeated measurements from the same subject over time), the only option to do trend analysis is to use an aggregate data to estimate the change over time at the level of a country. We have cited a reference that gives a more detailed explanation about our analysis (Line 119).

As a matter of rigour, while the use of "drivers" raises more interest it has a connotation of causality which this cross-sectional data cannot support. You have discussed this as a limitation and used the term "predictors". To be humble and acknowledge this limitation I suggest that you change the title and relevant wording in the text.   

Response: We preferred to use the term ‘driver’ instead of ‘predictor’ because the former gives the sense that we are evaluating the association for change over time (trend). The term predictor in its common use would give the sense that we are referring to a cross-sectional association at the individual level while we are interested in the association of the trend. We have also clearly mentioned the limitation with regard to causality in the discussion (Line 263). So we decided to keep the term ‘driver’.

Reviewer 3 Report

This is a generally well written report of an important analysis of the Demographic and Health Surveys since 2000 aimed at identifying drivers of the changes in stunting. The key results as indicated in table 4 are summarised in the abstract.

My only difficulty relates in trying to understand the differences between the two analytical models which result in the data reported in table 3 and table 4.

Table 3, Trends in stunting prevalence and potential determinants across survey-rounds (n = 50), is only referred to in relation to the overall average annual rate of stunting reduction (lines 162-164) and is as far as I understand the between country average annualized rate of change, the overall mean values obtained from the individual country data for each variable showed in Figure 1 and Supplementary data 1. This shows for example non significant changes in Prevalence of reported low birthweight, and significant changes in Prevalence of acute respiratory illness and Prevalence of diarrhea and this is consistent with a visual examination of the data in Supplementary data 1.

Table 4. (Modelling distal, intermediate, and proximal drivers of trend in stunting prevalence), is the main set of results  and is described as “the association between probability of stunting and the variations in potential  drivers within- and between-countries with the within-country association reported as a measure of the effect of an indicator on stunting trend over time” (lines 171-173). In this analysis the significance of some of the factors shown in table 3 is different. For example as proximal determinants of stunting Prevalence of reported low birthweight is now significant whereas acute respiratory illness and Prevalence of diarrhea are both non significant.

I am sure that as an informed reader of this paper I will not be alone in finding it difficult to reconcile these different findings. The authors should therefore include in their discussion an explanation of what the model used to generate the table 4 results involves and why these differences occur.

Author Response

This is a generally well written report of an important analysis of the Demographic and Health Surveys since 2000 aimed at identifying drivers of the changes in stunting. The key results as indicated in table 4 are summarised in the abstract.

My only difficulty relates in trying to understand the differences between the two analytical models which result in the data reported in table 3 and table 4.

Table 3, Trends in stunting prevalence and potential determinants across survey-rounds (n = 50), is only referred to in relation to the overall average annual rate of stunting reduction (lines 162-164) and is as far as I understand the between country average annualized rate of change, the overall mean values obtained from the individual country data for each variable showed in Figure 1 and Supplementary data 1. This shows for example non significant changes in Prevalence of reported low birthweight, and significant changes in Prevalence of acute respiratory illness and Prevalence of diarrhea and this is consistent with a visual examination of the data in Supplementary data 1.

Table 4. (Modelling distal, intermediate, and proximal drivers of trend in stunting prevalence), is the main set of results  and is described as “the association between probability of stunting and the variations in potential  drivers within- and between-countries with the within-country association reported as a measure of the effect of an indicator on stunting trend over time” (lines 171-173). In this analysis the significance of some of the factors shown in table 3 is different. For example as proximal determinants of stunting Prevalence of reported low birthweight is now significant whereas acute respiratory illness and Prevalence of diarrhea are both non significant.

I am sure that as an informed reader of this paper I will not be alone in finding it difficult to reconcile these different findings. The authors should therefore include in their discussion an explanation of what the model used to generate the table 4 results involves and why these differences occur.

Response: In the revised version of the manuscript, we have added a statement summarizing about the change over time in the indicator variables by stating that most indicator variables showed an improvement over time except some variables like coverage of women work opportunity (Line 165). It is true that the observed improvements were not statistically significant in some of the indicators like iron supplementation and reported low birth weight, which could be due to the wide confidence intervals because of small number of countries analyzed. Furthermore, we have also mentioned in the discussion that the lack of significant association between stunting trend and some of the indicator variables expected to be important drivers could be due to the lack substantial improvements in the indicators (Line 228 & Line 278-282).